# Squalamines in Blockade of Tumor-Associated Angiogenesis and Cancer Progression

**DOI:** 10.3390/cancers14205154

**Published:** 2022-10-21

**Authors:** Colin Sterling, Diana Márquez-Garbán, Jaydutt V. Vadgama, Richard J. Pietras

**Affiliations:** 1Division of Cancer Research and Training, Charles Drew University School of Medicine and Science, Los Angeles, CA 90059, USA; 2Division of Hematology-Oncology, Department of Medicine, UCLA David Geffen School of Medicine and UCLA Jonsson Comprehensive Cancer Center, Los Angeles, CA 90095, USA

**Keywords:** malignancy, squalamine, tumor-associated angiogenesis, vascular endothelial growth factor (VEGF), aminosterol, MAPK, FAK

## Abstract

**Simple Summary:**

Despite the many ongoing advances in cancer prevention, diagnosis and clinical management, global mortality from human cancers remains at high levels. Notably, more effective treatment of many cancers is advancing due to introduction of novel biologic therapies targeted to tumor signaling and immunologic pathways. Tumor growth is dependent on a sustained blood supply of nutrients and oxygen, and this process termed tumor-associated angiogenesis (TAA) has prognostic and therapeutic importance in several human malignancies. This review addresses use of squalamines to stop tumor growth. This naturally occurring compound can inhibit angiogenesis in tumors thereby reducing malignant progression in preclinical studies and in early clinical trials.

**Abstract:**

Mechanisms of action of squalamine in human vascular endothelial cells indicate that this compound attaches to cell membranes, potentially interacting with calmodulin, Na^+^/H^+^ exchanger isoform NHE3 and other signaling pathways involved in the angiogenic process. Thus, squalamine elicits blockade of VEGF-induced endothelial tube-like formation in vitro. Further, squalamine reduces growth of several preclinical models of human cancers in vivo and acts to stop metastatic tumor spread, actions due largely to blockade of angiogenesis induced by the tumor and tumor microenvironment. Squalamine in Phase I/II trials, alone or combined with standard care, shows promising antitumor activity with limited side-effects in patients with advanced solid cancers. Increased attention on squalamine regulation of signaling pathways with or without combination treatments in solid malignancies deserves further study.

## 1. Introduction

Cancer accounted for nearly 1 in 6 deaths worldwide in 2020. Further, it is suggested that the number of cancer cases will exceed 29 million, with the number of cancer deaths to surpass 16.3 million globally by 2040 [1]. The most common solid cancers include malignancies of the breast, lung and bronchus, prostate, colon and rectum, skin, kidney and bladder, endometrium and ovary, thyroid, pancreas and liver. Despite recent breakthroughs in the areas of surgery, chemotherapy, radiotherapy and immunotherapy that have helped to improve patient outcomes, it remains urgent to discover better antitumor strategies with enhanced antitumor efficacy and safety for use in the clinic. The effective management of cancer going forward will require more understanding of cellular and molecular signaling networks that modulate cancer growth and spread, both in the tumor mass and in the tumor microenvironment (TME). A main goal of this review is to summarize a host of publications that provide preclinical and clinical evidence for a role of squalamines in blocking tumor-associated angiogenesis and the progression of solid malignancies.

Rapid growth of tumors requires a sufficient supply of oxygen and nutrients. This process to create new blood vessels is referred to as angiogenesis and it has been described as an important predictive factor for human tumor progression [2,3,4,5]. This process of angiogenesis is controlled in part -by modulation of naturally occurring growth factors in the TME. Several growth factors are implicated in sustaining malignant growth, with vascular endothelial growth (VEGF) often a primary driver of tumor-associated angiogenesis. Additional growth factors, including fibroblast (FGF) and hepatocyte factor, also promote angiogenesis and consequent tumor cell proliferation in the TME. Further, hypoxia inducible factor 1 (HIF-1), a transcription factor induced by hypoxic conditions, is reported to increase the rate of endothelial cell proliferation in ischemic zones in tumors. Angiopoietins are yet another subset of growth factors that regulate vascular endothelial cell survival as well as interactions with supporting pericytes as detailed elsewhere [6].

In general, the dominant growth factor regulating tumor-associated angiogenesis is VEGF, and VEGF-A is the leading therapeutic-targeted member of the VEGF family for management of pathologic angiogenesis [6,7]. Hypoxia-inducible transcription factors (HIFs) are known regulators of VEGF expression by increasing VEGF gene transcription [8]. Most human solid tumors show upregulation of VEGF transcripts and protein that are produced by tumor cells, stromal cells, and endothelial cells. Importantly, VEGFs mediate mitogenic, pro-angiogenic signals to endothelial cells via high affinity receptor tyrosine kinases [7,9]. Receptors for VEGF (VEGFR) are expressed on both tumor endothelium and tumor cells and VEGFs act through three different VEGF receptors VEGFR1 (Flt1), VEGFR2 (Flk 1) and VEGFR3 (Flt4). Activation of tyrosine kinase receptors by VEGF binding induces homo- or heterodimerization and subsequent activation of downstream signaling pathways such as mitogen-activated protein kinase (MAPK), p38MAPK, phosphoinositide 3′ kinase (PI3K) and AKT [10,11,12]. Evidence has demonstrated VEGF overexpression is associated with progression of many cancers [13,14,15]. Common growth factor receptor pathways present in cancer cells, including HER2, IGF1 and EGF, can also contribute to upregulation of VEGF production and consequently increased angiogenesis [16,17]. EGF and HER family receptors play a key role in many human cancers [18,19] and overexpression of these receptors has been correlated with an increase in secretion of VEGF and stimulation of blood vessel formation [20,21,22].

The induction of angiogenesis is an important mechanism by which tumors promote their continuous growth and spread. As tumor cells proliferate, cells located more centrally in the growing tumor mass become hypoxic, thereby triggering release of pro-angiogenic factors to promote neovascularization [23,24]. Inhibition of tumor xenograft growth in preclinical studies was first reported by selective inhibition of VEGF by use of anti-VEGF monoclonal antibodies and by VEGF receptor tyrosine kinase inhibitors (TKIs) [25]. Bevacizumab (Avastin), a VEGF A-targeting monoclonal antibody, was first used in the clinic almost two decades ago, and it was one of the first targeted antiangiogenic therapies. This was the first time that an anti-angiogenic therapy was demonstrated to be effective in the clinic by prolonging the survival of patients with advanced colorectal cancer [26]. Clinical indications for management with bevacizumab now include colorectal cancer, ovarian cancer, breast cancer, non-small-cell lung cancer, glioblastoma, renal cell carcinoma and hepatocellular cancer [27,28,29,30]. In a review of clinical studies of bevacizumab therapy in patients with solid tumors, the median duration of bevacizumab treatment was 5 years [31]. Adverse events including proteinuria and hypertension were reported most often. However, resistance to bevacizumab in the clinic can arise by up-regulation in tumors of alternate pro-angiogenic growth factors that are not targets of bevacizumab [32].

In contrast to monoclonal antibodies, tyrosine kinase inhibitors (TKIs) targeting VEGFRs are orally bioavailable yet some lack specificity by inhibiting other tyrosine receptors and kinases involved in tumor signaling. Consequently, TKIs can exhibit toxicities due to off-target effects on non-VEGFR tyrosine kinases. Importantly, currently available antiangiogenic therapeutics, including TKIs and monoclonal antibodies, are generally reported to elicit a variety of cardiovascular [33,34] and non-cardiovascular [35] toxicities, including risks for proteinuria, hypertension and hemorrhage, likely related to effects on non-tumor vasculature. Among other approaches to blockade of tumor angiogenesis in preclinical development or in clinical use are inhibition or stabilization of hypoxia-inducible factors (HIF) [36], inhibition of TIE2/angiopoietin [37], and immunomodulatory drugs such as thalidomide and analogues [38]. Despite the current FDA-approved use of antiangiogenic drugs, inhibition of VEGF signaling has not proven to be effective in all tumor types. Success of antiangiogenic therapies has been relatively limited, often yielding only short-term relief and requiring prolonged use of anti-VEGF therapies. The limited efficacy of these antiangiogenic therapeutics may well be due to solid malignancies evolving alternate modes of blood vessel supplies and acquired resistance due to interactions among different components of the TME [32].

Recent preclinical and clinical trial findings have now elucidated another important role of antiangiogenic agents—namely in the modulation of antitumor immunity—and highlight the promise for utilizing combinations of angiogenesis inhibitors with immune checkpoint inhibitors (ICIs) [39,40,41,42,43,44]. Clinical use of ICIs, including programmed cell death 1 (PD-1), programmed cell death-ligand 1 (PD-L1), and cytotoxic T-lymphocyte-associated protein 4 (CTLA-4) inhibitors, may help to block checkpoint proteins, thus allowing T cell activation and killing of tumors. ICI therapy alone and/or combined with selected chemotherapies has led to durable clinical responses in patients with several different cancers including melanoma, non-small cell lung, triple-negative breast and renal cancers. However, most patients with cancer do not benefit from these treatments due in part to primary or acquired drug resistance to ICIs. The results of recent clinical trials provide evidence that use of ICIs combined with antiangiogenic therapeutics could be a promising new strategy to overcome the limited antitumor efficacy of ICIs, allow broader use of ICIs and improve patient survival. Notably, in addition to the critical role of VEGF in stimulating growth of the tumor vasculature, VEGF also acts to promote tumor immune evasion and progression. VEGF elicits disruption of T-cell function and increases the recruitment of regulatory T-cells and myeloid-derived suppressor cells (MDSCs), while simultaneously disrupting differentiation and activation of dendritic cells. Importantly, the TME comprises numerous immune, stromal and vascular cells, signaling factors and pathways that intersect with the angiogenic response. In turn, effective immunotherapy depends on increasing the presence of activated immune effector cells in the TME while also regulating the angiogenesis process [41]. Thus, combination of antiangiogenic agents with immunotherapies such as ICIs can potentially reverse an immunosuppressive TME to improve the anticancer efficacy of ICIs. These immunomodulatory properties of antiangiogenic therapeutics such as bevacizumab have led to new combination therapy approaches. Specifically, combination of cancer immunotherapy with bevacizumab has demonstrated clinical benefit in non-small cell lung cancer and for treatment of hepatocellular carcinoma [30,45] and renal cell carcinoma [46]. The potential for utilization of other types of antiangiogenic drugs in combination with immunotherapies remains to be investigated (see sections below).

## 2. Squalamines for Inhibition of Tumor-Associated Angiogenesis

As noted above, a main goal of this review is to summarize a host of scientific reports that provide preclinical and clinical evidence for a role of squalamines in blocking tumor-associated angiogenesis and the progression of solid malignancies (see Table 1). Squalamine is a naturally occurring steroid-polyamine conjugate compound initially reported to occur in some tissues of the dogfish shark, *Squalus acanthias* [47,48], and more recently in plasma membranes of white blood cells in the sea lamprey, *Petromyzon marinus* [49]. This compound is composed of a steroidal backbone structure akin to cholesterol, bearing a sulfated side chain and a hydrophilic polyamine spermidine group bound to a hydrophobic unit at C3 [48]. Squalamine induces changes in the shape of vascular endothelial cells and exhibits significant inhibition of angiogenesis in different human models of cancer [50,51,52,53]. Notably, squalamine, as well as a few other anti-VEGF therapeutics, is also reported to have beneficial effects as a therapy for management of neovascular age-related macular degeneration and potentially diabetic macular edema, see references [54,55,56].

In the shark, squalamine was found to occur primarily in liver and gallbladder [57]. Chemical synthesis of squalamine from other key intermediates and precursors including microbial metabolites has been published elsewhere [61,62,63]. Additional derivatives of squalamine such as des-squalamine and α-squalamine have also been described [64], and other aminosterols similar to squalamine were isolated from dogfish shark liver [65], but the potential antiangiogenic activity of these compounds has not been reported to date. Squalamine also possess antimicrobial, anti-fungal and anti-protozoa properties. More importantly, topical squalamine (0.2%) in combination with ranibizumab (0.5 mg) for ten weeks markedly restricted the formation of new blood vessels in patients with macular edema secondary to retinal vein occlusion. Please refer to reference [56]. Refer to Figure 1 for a display of squalamine structure.

In vitro, squalamine inhibits the growth of vascular endothelial cells, as evidenced initially by results of the chick embryo chorioallantois membrane assay. This finding was affirmed by rabbit corneal micro pocket assays and in vivo studies indicating inhibition of tumor-associated angiogenesis in gliomas and non-small cell lung xenografts in preclinical models [50,52,66]. Recent studies further show that squalamine can be taken up specifically by vascular endothelial cells and remain intracellularly for about 5 days [47]. Notably, squalamine inhibits the actions of VEGF and FGF, thereby blocking vascular endothelial cell growth and migration [51,66]. It has been postulated that several intracellular interactions with squalamine may involve interactions with endothelial Na^+^/H^+^ exchanger isoform NHE3, calmodulin or additional mechanisms localized in membrane structures [51,53,57], to block subsequent phosphorylation of kinases in blood endothelial cells such as p44/p42 mitogen-activated protein kinase (MAPK), protein kinase 2/p38 SAPK2 and focal adhesion kinase (FAK) [60]. A summary of key actions of squalamine in vitro and in vivo is shown in Table 1, with more detailed reviews below.

The strong antiangiogenic activity of squalamine is considered to result at least in part from its reported reversible inhibition of mammalian membrane bound Na^+^ exchange. This adjusts the intracellular pH and curbs growth factor mediated intracellular signaling, as well as additional downstream signaling pathways in endothelial cells [51,53,57]. As squalamine was initially reported to induce changes in cell shape, early research tested the hypothesis that selected transport proteins were targeted by squalamine. Since the Na^+^/H^+^ exchanger present in renal and intestinal brush borders is a transport protein that regulates cell volume as well as shape, Akhter et al. studied effects of squalamine on cloned mammalian Na^+^/H^+^ exchange isoforms, NHE1, NHE2 and NHE3, stably transfected in fibroblasts. In studies using transfected fibroblasts, repressive properties of squalamine on Na^+^/H^+^ exchanger activity were found to be time- and dose- dependent and reversible. This effect of squalamine is noted to have specificity for NHE3. By observing extracellular lactate dehydrogenase release, it was affirmed that squalamine did not have cytotoxic effects. Notably, Chen et al. implicated intracellular interactions between squalamine and calmodulin [67]. Using squalamine conjugated to a fluorescence marker, human vascular endothelial cells showed a rapid re-localization of squalamine and calmodulin from the cell periphery to perinuclear endosomal compartments. This was accompanied by disruption of F-actin stress fibers and significant changes in endothelial cell shape. Moreover, squalamine was reported to impede secretion of protons by mitogen-stimulated vascular endothelial cells, coinciding with previous results by Akhter et al. [52,57]. Accordingly, squalamine modulation of intracellular pH in vascular endothelial cells potentially by interactions with calmodulin could have a major impact on intracellular signaling by endothelial growth factors such as VEGF.

In studies using rat brain endothelial cells, squalamine was reported to inhibit mitogen mediated proliferation and migration, but this effect only occurred in cells stimulated by mitogenic factors [52]. This study further involved treatment of 4-day-old chick embryos with squalamine. Addition of squalamine had a direct effect on the vasculature after about 20 min by inducing constriction of the smaller capillaries and blockage of the circulation of red blood cells. Furthermore, persistent blood flow via larger vessels was altered and expanded to move through once-obstructed vessels. The net effect was a reduction of flow of blood through an altered capillary network. With these vessels predominantly consisting of endothelial cells, researchers suggested that squalamine-induced changes in endothelial cell remodeling. Further studies also proposed that systemic treatment of rats with squalamine led to suspended growth of rat flank 9 L gliomas due to a reduction in the density of blood vessels present in the tumor [52].

Williams et al. also reported that stimulation of HUVECs with VEGF followed by treatment with squalamine, resulted in disruption of F-actin polymerization of fibers in HUVECs [58]. Further, after squalamine treatment, significantly less vascular endothelial (VE)-cadherin was noticeable at endothelial cell–cell junctions, with increments in perinuclear distribution of VE-cadherin. Squalamine promoted internalization of VE-cadherin from endothelial surface membranes into the intracellular space. Notably, VE-cadherin plays a vital role that is integral to sustaining cell–cell adherence junction, and it is also linked to actin filament attachment sites on the vascular endothelial cell surface. Márquez-Garbán et al. reported more recently that squalamine stopped the growth of HUVECs by reduction of endothelial tube-like formations induced by VEGF in vitro [60]. Some of the actions of squalamine included blockade of focal adhesion kinase (FAK) phosphorylation and inhibition of cytoskeletal organization in HUVECs (Figure 2). Thus, squalamine can block endothelial cells from forming new blood vessels required by tumors to continue growing.

## 3. Preclinical Study of Tumor Cell Lines with Squalamine Treatment

Antitumor properties of squalamine administered alone or with selected chemotherapies were investigated in human and murine lung cancer xenograft models [68]. Targeted cell lines in the study included: H460 large cell carcinoma, NL20T-A and mouse Lewis lung carcinoma. Notably, squalamine used as a single agent or in combination with cisplatin had no cytotoxic effect in vitro among these cell lines. In contrast, using preclinical xenograft models in vivo, inhibition of tumor formation by squalamine alone was observed only when xenograft models were treated with squalamine before, simultaneously, or within 24 h following implants of tumors in mice. Squalamine combined with chemotherapy treatment elicited a significant inhibition in tumor xenograft growth in all cell lines tested. However, no additive or synergistic antitumor activity was detected in squalamine groups treated with paclitaxel, vinorelbine, gemcitabine or docetaxel chemotherapies. Actions of squalamine on the tumor vasculature were also assessed by immunohistochemistry (IHC) in H460 tumor xenograft samples post-treatment. Squalamine alone did not alter or influence the numbers of tumor-associated CD31-positive vessels. Notably, squalamine-cisplatin treatments did reduce by 25% the number of tumor associated CD31-positive vessels. The results indicated that squalamine was most effective when used in combination with platinum-based analogues, with antiangiogenic efficacy optimal when the drugs were administered simultaneously.

Using other NSCLC models, Williams and colleagues [58] observed that squalamine administered either alone or in combination with either cisplatin or carboplatin plus paclitaxel reduced tumor growth. MV-522 tumors were implanted as xenografts in immunodeficient mice. Thereafter, squalamine was given alone or combined with cisplatin or as part of triplet therapy with paclitaxel plus carboplatin. The results showed that squalamine with cisplatin or a cocktail triplet treatment with carboplatin and paclitaxel had greater efficacy than squalamine administered as a single agent in blocking the progression of MV-522 human lung tumors. After excision, xenograft tumors treated with the cocktail therapy weighed approximately 557 mg less than squalamine treated xenografts. These promising results led to further investigation of squalamine particularly in combination with selected chemotherapies in advanced lung cancer models.

Independent reports further suggest that ionized radiation in combination with antiangiogenic agents ameliorate tumor elimination while feasibly avoiding adverse side-effects [69]. Accordingly, the effects of squalamine on human non-small cell lung tumor H23 xenograft progression in nude mice were assessed with squalamine treatment alone and in combination with radiation therapy [70]. Squalamine unaccompanied with other treatments elicited modest efficacy in preventing H23 tumor growth. However, combination of squalamine with radiation therapy, commonly used for some non-small cell lung cancers, was significantly more effective in suppressing tumor growth. This suggests that combination of radiation therapy with squalamine may also prove effective in limiting tumor progression.

An important set of studies on the potential role of squalamine in preventing lung cancer spread was reported by Teicher et al. [50]. To determine antitumor efficacy of squalamine administered alone or in combination with cytotoxic chemotherapies in a model of primary and metastatic disease, squalamine was administered by daily subcutaneous injection or by continuous infusion on days 4–18 after implantation of Lewis lung carcinomas in preclinical animal models. Squalamine used as a single agent had modest effects on the growth of primary Lewis lung tumors but significantly enhanced tumor growth delays elicited by either cyclophosphamide, cisplatin, paclitaxel or 5-fluorouracil treatments by 2.4- to 3.8-fold compared with the chemotherapy drugs administered alone. Importantly, squalamine given alone markedly decreased the number of lung metastases in mice bearing Lewis lung carcinomas and further reduced the numbers of lung metastases when used in combination with chemotherapeutic drugs. After 20 days, metastases were reduced by half as compared to appropriate controls. Since lung metastases actively spread and grow in part due to formation of new blood vasculature, this unanticipated anti-metastatic action of squalamine highlights its significant antiangiogenic/antitumor potency in vivo. Given the poor prognosis of patients presenting with metastatic disease, further follow-up and confirmation of these promising results are clearly warranted.

Potential effects of squalamine on breast cancer progression post-chemotherapy treatment was studied using MX-1 breast xenografts grown in mouse models [71]. After tumor regression post-chemotherapy with cyclophosphamide, mice were randomized into different treatment groups. Two different doses of squalamine (10 or 20 mg/kg/day) therapy resulted in complete regression of a significant number of tumors. Similarly, studies by Teicher et al. [50] revealed that squalamine reduced the growth rate of rat 13,762 mammary carcinomas when treated with 40 mg/kg of squalamine; while squalamine in combination with chemotherapeutics reduced growth rate by 1.9–2.5-fold when agents were used together.

More recently, Márquez-Garbán et al. [60] reported on the influence of squalamine alone or with trastuzumab on MCF-7 breast tumor cell xenograft growth in nude mice without (MCF-7) or with HER-2 oncogene overexpression (MCF-7/HER-2). It is important to note that HER-2/neu overexpression is prevalent in 25–30% of breast cancers. Use of trastuzumab (Herceptin) therapy significantly blocks tumor growth in cancers with HER-2 overexpression [72,73]. Notably, VEGF secretion is enriched by HER-2 overexpression. Squalamine treatment of human MCF-7 breast tumor xenografts without HER2 overexpression readily inhibited cancer progression, while squalamine and trastuzumab as a combination therapy elicited marked inhibition of MCF-7/HER2 tumor growth surpassing the antitumor efficacy of trastuzumab treatments used alone. Further experiments also showed that squalamine inhibited human HUVEC growth and markedly reduced VEGF-mediated endothelial tube-like formations in vitro. Additional studies suggested that these cellular effects are associated with the blockade of FAK phosphorylation and stress fiber assembly in HUVECs. Thus, squalamine was able to effectively inhibit growth of breast cancer tumors with or without HER-2-overexpression by blocking tumor-associated angiogenesis.

VEGF also plays an important role in progression of ovarian cancer as has been demonstrated before [16]. Despite advancements in surgery and chemotherapy, overall 5-year survival rates for ovarian cancer have remained around 49%, making ovarian cancer the deadliest malignancy of gynecologic origin [74,75,76,77]. Indeed, tumor-associated angiogenesis has prognostic significance in epithelial ovarian cancer [13]. VEGF is heavily implicated in prognostic outcomes for epithelial ovarian cancer [13,14,15]. VEGF plays a key role in increasing vascular permeability [2,78,79], as well as its role in the formation of malignant ascites in ovarian cancer patients [80]. Based on the findings above, squalamine’s inhibitory effect on ovarian cancer was assessed by Li et al. [16]. Ovarian cancer cells were reported to secrete elevated levels of VEGF, but squalamine when used alone had no effect on tumor cell VEGF secretion, nor did it suspend growth of ovarian cancer cells grown in vitro. However, squalamine at 160 nanomolar doses halted proliferation of human vascular endothelial cells. Remarkably, at such minute doses, squalamine significantly reduced VEGF-induced capillary tube-like formations by the endothelial cells growing in 3D cell cultures [81]. Downstream, squalamine induced inhibition of rapid VEGF-induced activation of p44/p42 MAP kinase in vascular endothelial cells, an early response leading to activation of proliferation. Importantly, squalamine also inhibited VEGF-induced activation of FAK and protein kinase-2/p38 SAPK2, blocking, in turn, assembly of F-actin stress fibers in the endothelial cells, similar to findings on squalamine action in independent studies [51]. These effects followed the primary interaction of squalamine with caveoli at the surface membranes of vascular endothelial cells, sites well known to concentrate key signaling complexes to regulate the process of angiogenesis [12]. Molecular actions of squalamine to disrupt intracellular interactions downstream in vascular endothelial cells may well be integral in curbing the process of TAA. Finally, Li et al. [16] also assessed the antitumor efficacy of combined treatment with squalamine and platinum-based chemotherapies in two different human ovarian cancer cell lines with and without HER-2 overexpression grown as xenografts in immunodeficient mice. Among the several ovarian cancer models, combined treatment with squalamine and the platinum-based chemotherapies (for example, cisplatin or carboplatin) elicited profound tumor growth inhibition. Following squalamine treatment, tumor growth was restrained for 18 days. This in vivo study further showed that combined squalamine-chemotherapy optimally hindered tumor progression treatment. Ovarian tumors harvested from treated mice were later assessed for tumor cell apoptosis. In vivo studies with squalamine suggest that squalamine induces tumor cell apoptosis by increased cytotoxicity when combined with cisplatin facilitating tumor cell apoptosis. Furthermore, IHC studies confirm that TAA is reduced across each of the four different types of ovarian tumors.

Reports by Sokoloff et al., (2004) indicate that androgens and androgen deprivation treatments can also modulate angiogenesis in prostate cancer [82,83,84,85]. Androgen deprivation promotes prostate cancer cell apoptosis while concurrently reducing VEGF and increasing flt-1, a soluble short protein alternative transcript of VEGFR-1 that is expressed on endothelial and macrophage cells and is known to promote metastasis [86]. Notably, squalamine administered in combination with castration (androgen deprivation) led to further marked regression of prostate tumor xenografts in preclinical models [83,85,87]. However, squalamine alone was not effective in stopping the progression of prostate tumor xenografts in vivo. The investigators postulate that the castration-induced increase in flt-1 expression in the presence of squalamine enhances in turn inhibition of the expression of integrins v3 and v5, thereby promoting tumor regression. Based on their preclinical studies, the investigators suggested that clinical trials using a combination of androgen ablation and squalamine therapy should be planned for men determined to be at high risk for tumor recurrence after undergoing radical prostatectomy [83].

It is notable that both estrogen and growth factor receptor (HER-2 and EGFR) signaling pathways are also reported to regulate the secretion of VEGF that in turn stimulates tumor-associated angiogenesis. As in prostate cancer, these molecular interactions can significantly impact breast cancer progression, and integration of anti-angiogenesis agents such as squalamine with signal transduction inhibitors or antiestrogen therapies may lead to new antitumor treatment strategies [88]. More recently, Carmona et al. (2019) [89] developed a derivative of squalamine termed NV669, a polyaminosteroid compound, that was reported to have significant antitumor effects in pancreatic and hepatic cancer cells. Results showed that NV669 reduced cancer cell viability, elicited cell cycle arrest and stimulated apoptosis in vitro. These actions were accompanied by a reduction in cyclin B1 expression and phosphorylated CDK1 and by cleavage of pro-apoptotic caspase-8 and PARP-1. NV669 also inhibited PTP1B activity and FAK expression and altered expression of several adhesion molecules, leading to tumor cell detachment and apoptosis. Furthermore, in pancreatic and hepatic cancer cell xenografts in nude mouse preclinical models, about 75% of tumors from NV669-treated mice had significantly lower tumor volumes than control tumors, and there was no NV69 cytotoxicity as assessed by the lack of changes in animal body weight and behaviors during the course of the studies. To determine if NV669 as squalamine elicited any antiangiogenic effects, mean numbers of CD31-positive capillary tube formations per microscopic field were assessed by use of IHC methods. The results showed that the mean numbers of capillary tube formations were reduced in NV669-treated tumors as compared to controls as reported before [54], but the difference was not statistically significant. However, NV669 therapy did promote significant cell cycle arrest in the pre-mitotic phase and increased tumor cell apoptosis. These findings indicate that this new derivative of squalamine appears to have some biologic actions similar to that of squalamine [54,56], but it also exhibits additional antitumor actions that may be distinct from parental squalamine. As noted above, several additional chemical analogues of squalamine and related steroid polyamines have been reported previously [90,91], but antiangiogenic and/or antitumor activity of these derivatives has not been established to date.

## 4. Clinical Trial Studies with Squalamine

Based on promising preclinical data around various cancer models, squalamine, was advanced for clinical development as a potential therapeutic to treat human malignancies. Squalamine used in clinical trials was synthesized in approved pharmaceutical facilities [61]. Bhargava et al. undertook a comprehensive clinical study of squalamine’s activity in patients with advanced solid cancers in Phase I clinical trials [92]. The cohort included 19 patients that had Eastern Cooperative Oncology Group (ECOG) performance scores ≤2 with advanced non-leukemic cancers and significant disease burden. After a 5-day continuous i.v. treatment of squalamine, patients optimally tolerated a dose of 192 mg/m^2^/day; however, these studies also revealed that 384 mg/m^2^/day was also relatively well-tolerated by study subjects. The dose-limiting toxicity (DLT) noted with squalamine treatment was hepatotoxicity, but this finding was transient and reversible. Mild to moderate fatigue occurred in some patients, but this resolved post-treatment. Pharmacokinetic studies indicated that squalamine has a relatively high plasma clearance and maintained a short elimination half-life initially. In this Phase 1 trial, squalamine was well-tolerated by patients with an approximate dose rate around 12–384 mg/m^2^/day.

Hao et al. reported an independent Phase I trial [93]. Patients (n = 33) with advanced solid cancers were treated with a 5 day i.v treatment every three weeks. As in the Bhargava et al. [92] trial, the main DLT was elevated hepatic transaminases and hyperbilirubinemia, effects that were again reported to be transient, asymptomatic and reversible after the stop of treatment. Based on the DLT findings, the dose of squalamine was recommended not to surpass 500 mg/m^2^/day as a continuous i.v. infusion over 5 days per 3 weeks. In this study, the pharmacokinetics for squalamine exhibited dose-proportional kinetics, whereas Bhargava et al. [92] using a narrower dosing range determined that the AUC_0–t_ and C_max_ were nonlinear at the highest doses. After 12–15-weeks of therapy, some patients presented with evidence of stable disease, but no major tumor regression was recorded. Researchers recommended a Phase II dose of 500 mg/m^2^/day, as this dose was well-tolerated, with optimal antiangiogenic effects.

Since several preclinical studies revealed that squalamine had harmonious or coactive antitumor activity with selected chemotherapies, Herbst et al. [59] led a Phase I/IIA clinical trial to assess the success of squalamine combined with paclitaxel and carboplatin in patients with advanced NSCLC. This comprehensive study assessed the clinical performance, as well as the initial toxicity profile of squalamine when administered by infusion over the course of 5 days in combination with standard chemotherapy in patients with advanced stage IIIB or stage IV NSCLC. The suggested MTD for the Phase II arm of the trial was found to be 300 mg/m^2^/day. This dose was used for up to six 21-day cycles. Notably, toxicities in this study (in which squalamine was administered at lower dose levels than in prior Phase I trials) were limited to only mild myelosuppression. The trial showed that squalamine had linear pharmacokinetics (PK), and clearance of squalamine in plasma did not change at this dosage. In addition, the PK profile of paclitaxel was unchanged by combining it with squalamine, nor was carboplatin drug clearance altered by co-treatment with squalamine. Of standard clinical endpoints to assess antitumor efficacy, partial clinical responses (PR) but no complete responses were recorded, with an overall response rate (ORR) of 28% among evaluable patients. Stable disease occurred in 19% of patients, while 53% of patients had disease progression. Thus, a significant clinical benefit rate was found for 47% of evaluable patients with advanced NSCLC. When treated with squalamine MTD of 300 mg/m^2^/day, median patient survival was 10 months, and 1-year survival was 40%. Thus, the improved patient survival data as compared to historical trial data using paclitaxel-carboplatin regimens [94], and the reported safety profile suggests that the use of squalamine with selected chemotherapies shows exceptional promise for patients with advanced stage NSCLC.

A phase II clinical trial was initiated by Davidson et al. to assess the properties of squalamine and actions of squalamine in the treatment of human ovarian cancer [95]. In this trial, 33 patients that had Stage 3–4 chemotherapy-resistant or refractory ovarian cancer received a 5-day continuous infusion of carboplatin along with squalamine (200 mg/m^2^/day) over the course of 81 days. Of 22 evaluable patients, 8 patients (36%) achieved an objective response. Three patients exhibited toxicities which included anemia, leukopenia and additional symptoms commonly associated with carboplatin chemotherapy. To date, no additional follow-up data are available. Nonetheless, the results above suggest that further studies should be considered to explore the potential role of squalamine as a noteworthy candidate to supplement existing treatments for patients with advanced and/or refractory ovarian cancers. Notably, the US Food and Drug Administration (FDA) previously granted orphan drug status for squalamine to advance its development as a new antitumor agent to treat resistant or refractory ovarian cancer.

## 5. Conclusions

As noted by Lugano et al. [32], the overall success to date of anti-angiogenic therapy alone for malignancies has been modest. The limited antitumor efficacy may be due to tumor deployment of alternative vascularization mechanisms and/or resistance to current therapies. Squalamine, a naturally occurring cationic steroid, elicits modifications in vascular endothelial cell shape and function, and has been shown to evoke marked inhibition of angiogenesis in preclinical studies of lung, breast, prostate, brain and ovarian cancers. The mechanism of squalamine activity involves the inhibition of endothelial cell proliferation and migration stimulated by several different known mitogens [47,52]. Tumor xenograft studies using in vivo models with squalamine have shown that it has antitumor activity as a single agent in several but not all models and is more effective in blocking tumor progression when combined with either chemotherapies or radiation therapy. Importantly, preclinical studies with lung carcinoma models further demonstrated that squalamine treatment significantly blocked tumor metastasis, thereby affirming a strong antiangiogenic capability of squalamine. In early stage clinical trials, squalamine displays relatively low systematic toxicity amongst selected patient populations [92,93,95]. Accordingly, these collective findings warrant further investigation of squalamine and the launch of future randomized, controlled clinical trials of the antitumor efficacy of squalamine or structural analogues with more potent antiangiogenic and antitumor activity and/or with improved bioavailability [47]. Thus, squalamines administered in combination with chemotherapy, radiotherapy or potentially immunotherapies as noted above indicate the potential for this naturally occurring therapeutics to be a valuable addition to the clinical armamentarium available for patients afflicted with solid tumor malignancies [31,43,44].

## Figures and Tables

**Figure 1 cancers-14-05154-f001:**
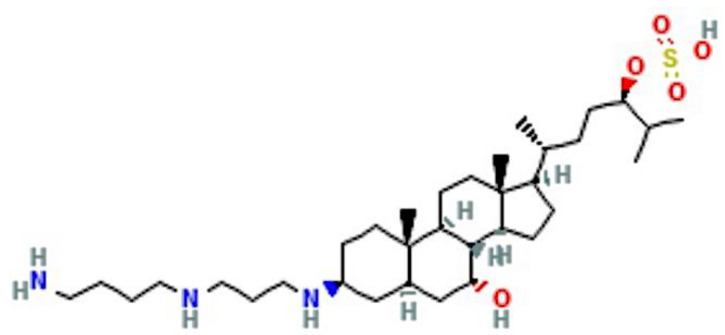
Squalamine, a 7,24 dihydroxylated 24-sulfated cholestane steroid conjugated to a spermidine at position C-3 (C_34_H_65_N_3_O_5_S). The molecular weight of squalamine is 628 g/mol, see text for details and PubChem (https://pubchem.ncbi.nlm.nih.gov (accessed on 1 April 2022)).

**Figure 2 cancers-14-05154-f002:**
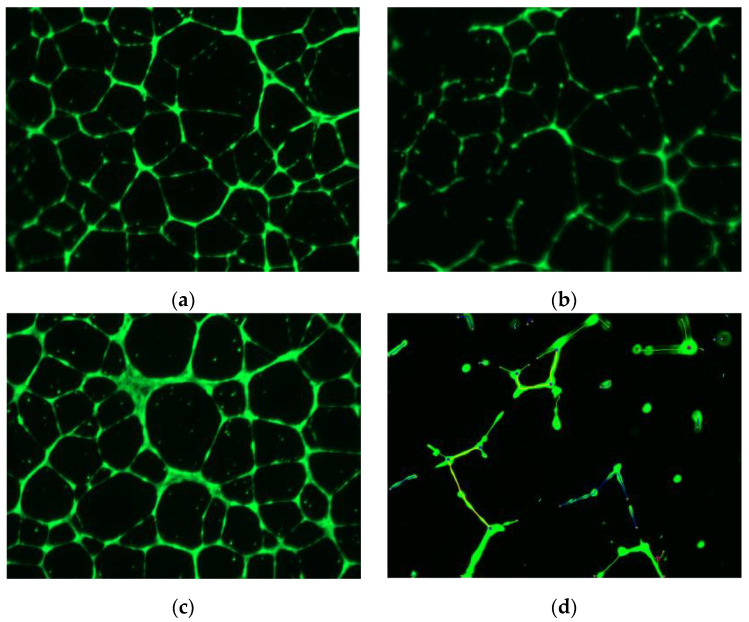
Squalamine inhibits capillary tube-like formations and the skeleton of tubular networks formed in an Endothelial Tube Formation Assay. Human umbilical vein endothelial cells (HUVEC) were grown in medium without serum on the surface of Geltrex (Thermofisher) in the presence of the following conditions: (**a**) control vehicle, (**b**) squalamine 1 μM; (**c**) VEGF 50 ng/mL and (**d**) VEGF (50 ng/mL) + squalamine 1 μM for 18–24 h. Tubular structures formed by endothelial cells were analyzed by fluorescent microscopy and photographed. Angiogenesis Analyzer ImageJ (Gilles Carpentier, Faculte des Sciences et Technologie, Universite Paris Est Creteil Val-de-Marne, France) was used to determine differences in structures. Pictures are representative of at least three different experiments [51].

**Table 1 cancers-14-05154-t001:** Summary of Key Findings on Squalamine Actions in vitro and in vivo.

* Development	Reference
Squalamine specifically inhibits cell surface sodium-proton transporter in fibroblast and ileum models in vitro	Akhter et al., 1999 [57]
(a) Squalamine disrupts F-actin filament organization and down-regulates expression of E-cadherin in preclinical models (b) Squalamine in combination with cisplatin increases response to chemotherapy in human lung tumors in nude mice.	Williams et al., 2001 [58]
(a) Squalamine inhibits VEGF-induced MAPK activation in HUVEC cells. (b) Combination of squalamine with cisplatin blocks growth of human ovarian cancer xenografts with or without HER-2 gene overexpression by inhibition of angiogenesis.	Li et al., 2002 [16]
Phase I/II clinical trial of 5-day infusion of squalamine lactate (MSI-1256F) in combination with paclitaxel and carboplatin in patients with advanced stage NSCLC, ORR was 28%, with stable disease in 19%.	Herbst et al., 2003 [59]
(a) Squalamine inhibits endothelial tube-like formation induced by VEGF in HUVEC cells that is mediated by FAK. (b) Squalamine blocks progression of human breast cancer xenografts with or without HER-2 overexpression by inhibiting angiogenesis.	Márquez-Garbán et al. 2019 [60]

* Squalamine has shown antiangiogenic/antitumor efficacy across multiple cancer cell types including ovarian, non-small cell lung cancer, breast and more recently, pancreatic, glioblastoma and hepatic cancer cell types. Ongoing research suggests that squalamine inhibits downstream activity of selected factors such as VEGF associated with cancer cell progression and growth. Squalamine is well-tolerated in Phase I-II clinical trials in combination with chemotherapeutic agents. VEGF (vascular endothelial growth factor); MAPK (mitogen activated protein kinase), HUVEC (human umbilical vein endothelial cells); HER-2 (human epidermal growth factor receptor 2); non-small cell lung cancer (NSCLC); ORR (overall response rate); FAK (Focal Adhesion Kinase).

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
