# Peer review of "Squalamines in Blockade of Tumor-Associated Angiogenesis and Cancer Progression"

_cancers, 2022, doi:10.3390/cancers14205154_

Round 1

Reviewer 1 Report

The article "Squalamines in blockade of tumor-associated angiogenesis and cancer progression" reviews the importance of squalamines in obstruction of angiogenesis and tumor progression. 

Major comments:

- The Squalamines are also utilized as  antibiotics as well as for eye infections! Do the authors think, what they conclude as modest anti-angiogenic therapy can be considered as an effective alternative over the available ones or in combinations?

- Many factors are reported to modulate cancer progression activities, recently bacteriophages have also been reported to affect signal transduction pathways in prostate cancers. What receptors might be involved and are the affecting cancer cell growth, migration, viability, etc temporarily or permanently must be explained further!

Sanmukh, S. G., Santos, N. J., Barquilha, C. N., Dos Santos, S., Duran, B., Delella, F. K., Moroz, A., Justulin, L. A., Carvalho, H. F., & Felisbino, S. L. (2021). Exposure to Bacteriophages T4 and M13 Increases Integrin Gene Expression and Impairs Migration of Human PC-3 Prostate Cancer Cells. Antibiotics (Basel, Switzerland), 10(10), 1202. https://doi.org/10.3390/antibiotics10101202

Sanmukh, S. G., & Felisbino, S. L. (2018). Development of pipette tip gap closure migration assay (s-ARU method) for studying semi-adherent cell lines. Cytotechnology, 70(6), 1685–1695. https://doi.org/10.1007/s10616-018-0245-1

Yang, C.; Bromma, K.; Chithrani, D. Peptide Mediated In Vivo Tumor Targeting of Nanoparticles through Optimization in Single and Multilayer In Vitro Cell Models. Cancers 2018, 10, 84. https://doi.org/10.3390/cancers10030084

- What are the limitations associated with Squalamines and are they considerably effective in-vivo.

Minor comments:

- Protein- protein interactions between over expressed and down regulated genes can be explained to show the effects Squalamines have on cancer progression processes like cell growth, migration and cell death.

- Previous work by the authors can be co-related with other reported studies.

Author Response

Reviewer #1

Comment: ‘The Squalamines are also utilized as antibiotics as well as for eye infections! Do the authors think, what they conclude as modest anti-angiogenic therapy can be considered as an effective alternative over the available ones or in combinations?’ 

RESPONSE: We concur with the reviewer. Squalamines are reported to have 
antibacterial properties (Nicol M, Int J Antimicrob Agents. 2019, Alkhzem AH, RSC Adv. 
2022 Douglas EJA, Front Microbiol. 2022) and are being assessed as potential 
treatments for retinal diseases such as retinal neovascularization (Kazakova OB, 
Molecules, 2020, Al-Khersan, Expert Opin Pharmacother. 2019). As we state in the review, bevacizumab has proven to exert significant antiangiogenic/antitumor activity in the clinic. However, use of bevacizumab may also be associated with some adverse events, and therapeutic resistance to bevacizumab may emerge by up-regulation of proangiogenic factors that are not targeted by bevacizumab (page 2, line 57-62). With further clinical translational development, squalamine or derivatives may ultimately offer an 
effective alternate treatment strategy, as squalamine acts by alternative mechanisms of action. It is also possible that squalamines may be utilized in combination with other antiangiogenic and/or antitumor therapies. The use of antiangiogenic therapies with immunotherapies is also a promising new direction of antitumor therapy (Fukumora et al., 2018; 2020; Song et al., 2020).

Comment:’ Many factors are reported to modulate cancer progression activities, recently bacteriophages have also been reported to affect signal transduction pathways in prostate cancers. What receptors might be involved and are the affecting cancer cell growth, migration, viability, etc temporarily or permanently must be explained further!’

RESPONSE: We agree with the reviewer that many factors contribute in the regulation of cancer progression. Several of these factors are detailed in the review (Ferrara N, 2003, Ellis LM, 2008, Hu CJ, 2003, Koch S, 2012, Rousseau S, 1997, 2000). Indeed, squalamines are among the potential therapeutics that are reported to block the progression of prostate cancer in preclinical models (Sokoloff et al., 1999; 2004). Mechanisms of squalamine action in preclinical models are reviewed in the manuscript and include squalamine disruption of downstream VEGF receptor signaling that leads to inhibition of integrin expression, induction of apoptosis and disruption of cytoskeletal formations that leads to inhibition of endothelial cell growth and migration. In addition, squalamine is reported to interact with mammalian membrane-bound Na+/H+ exchangers by reversible inhibition causing changes in cellular pH and inhibiting growth factor downstream signaling. The antiangiogenic action of squalamine appears to be time-limited in clinical trials, and sequential dosing of squalamine is needed (Bhargava P, 2001, Lugano R, 2020). We note that Mammari et al. (2022) have also recently reported on mechanisms of action of squalamine, and we include this reference in the revised manuscript.

As implied by the reviewer in the references cited, it may be that antibacterial 
properties of squalamine could modulate the bacterial microenvironment in the GI tract in vivo, thereby indirectly affecting tumor progression, but we are not aware of any published reports to date that provide evidence for this action of squalamine in vivo. This represents an important topic to be further studied. We note that another squalamine review recently published by Mammari et al. (2022) in the journal Microorganisms provides more detail about the antibacterial actions of squalamine.

Comment: ‘What are the limitations associated with Squalamines and are they 
considerably effective in-vivo.’

RESPONSE: As detailed in the review, squalamines have been assessed in preclinical models in vitro (pages 3,4,6) and in vivo (pages 3,5-7), as well as in Phase I-II clinical trials in patients with non-small cell lung cancer and ovarian cancer (pages 10). In preclinical models and in clinical trials, squalamine exhibited significant antiangiogenic and antitumor efficacy with limited, reversible side-effects. One limitation for in vivo use is the need to administer squalamine i.v. Overall, the findings indicate that further development of squlamines or derivatives for management of malignancies is feasible to 
pursue going forward (refer to Table 1).

Minor Comments

Comment: ‘Protein- protein interactions between over expressed and down regulated genes can be explained to show the effects Squalamines have on cancer progression processes like cell growth, migration, and cell death.’

RESPONSE: We concur with the reviewer that protein-protein interactions likely play an important role in the effects of squalamine on tumor-associated angiogenesis and cancer progression. We have discussed these interactions on pages 5,6 and 8 of the review.

Comment: ‘Previous work by the authors can be co-related with other reported studies.’

RESPONSE: As stated by the reviewer, the authors of this manuscript have contributed previous reports on the potential role of squalamine in modulating tumor-associated angiogenesis and cancer progression. A main goal of this review is to collate a host of publications on the role of squalamines in solid malignancies to potentially establish a way forward in the further development of this compound. We also cite in the revised manuscript a new report by Mammari et al. (2022) in support of this goal.

Concluding Remarks: After in depth review of the manuscript as guided by the reviewer critiques, please note that we have edited some spelling/grammatical errors in the original manuscript text that we corrected in the revised manuscript. These and other modifications requested by reviewers were changed using ‘track changes’ in the revised manuscript in order to accommodate the journal readers.

Reviewer 2 Report

The author has to emphasize on the reason for choosing squalamines as a topic for this review and need to state this reason before moving to the subtopic.

The author has to explain the cf abbreviations.

Were these squalamines discovered after bevacizumab for anti-angiogenesis treatment? If yes , what is the relationship between these two compounds?

In line 20 , the author has stated"  low doses of squalamine restricted the formation of new blood vessels", what is the low dose and do you have supplementary data supporting this statement?

What is the cytotoxic dose of Squalamine and it's IC150 value?

The paragraph about VEGF from line 79 can be moved to the introduction to better fit the paper. 

Which type of cancer has an effect when treated with Squalamines and what is the efficacy rate?

Author Response

Reviewer #2

Comment: ‘The author must emphasize on the reason for choosing squalamines as a topic for this review and need to state this reason before moving to the subtopic.

RESPONSE: In response to the reviewer’s recommendation, we revised the manuscript to state: “A main goal of this review is to summarize a host of publications that provide preclinical and clinical evidence for a role of squalamines in blocking tumor-associated angiogenesis and the progression of solid malignancies.” This change is noted in the document entitled “Track changes-squalamine manuscript” and visible in the “Final squalamine manuscript” attached.

Comment: ‘The author must explain the cf abbreviations.’

RESPONSE: We use the standard abbreviation ‘cf.’, a shorthand for the Latin word confer meaning ‘compare’. We used the abbreviation to welcome the reader to compare claims made in our review literature to the work cited by cf. We will replace ‘cf.’ with the term ‘refer to’ in the revised text to prevent any confusion. 

Comment: ‘Were these squalamines discovered after bevacizumab for anti-angiogenesis treatment? If yes, what is the relationship between these two compounds?’

RESPONSE: Published scientific reports about squalamines and about an effective antibody against VEGF occurred in 1993. Napoleon Ferrara and colleagues demonstrated that monoclonal antibodies were effective at targeting VEGF and inhibiting its downstream signaling (Kim K, Nature, 1993). Discovery and development of bevacizumab was then described (Ferrara NL, Nature Reviews, 1993). In 1993, the group led by Michael Zasloff reported that squalamine, an aminosterol isolated from the dogfish shark, had antibacterial properties against gram-positive and gram-negative bacteria (Moore KS, Proc Natl Acad Sci USA, 1993). However, the first report demonstrating inhibition of angiogenesis by squalamine did not occur until 1998 when Sills et al. demonstrated that squalamine inhibited angiogenesis in different animal models. This report indicated that this squalamine effect was mediated, at least in part, by blocking mitogen-induced proliferation and migration of endothelial cells, thus preventing neovascularization of tumors. Importantly, it was also reported that squalamine had no effect on normal endothelial cells. (Sills AK, Cancer Res, 1998). Thus, these two compounds are related in part in their ultimate mechanism of action - they both exert anticancer effects mediated by inhibition of tumor-associated angiogenesis. Squalamine has shown promise as an alternative option to bevacizumab, potentially with less side effects. 

Comment: ‘In line 20 , the author has stated" low doses of squalamine restricted the formation of new blood vessels", what is the low dose and do you have supplementary data supporting this statement?’ 

RESPONSE:

The Sentence “Low dose… in line 20” refers to reference 56 (Wroblewski, J.J. et al, 2016). This sentence has been modified in the manuscript to clarify the dosage of squalamine. Line 20 onward now reads: “More importantly, topical squalamine (0.2%) in combination with ranibizumab (0.5 mg) for ten weeks markedly restricted the formation of new blood vessels in patients with macular edema secondary to retinal vein occlusion. Please refer to reference [56].“ 

Comment: ’What is the cytotoxic dose of Squalamine and it's IC150 value?’

RESPONSE: In early phase clinical trials, dose-limiting toxicities (DLT) of squalamine varied somewhat depending on cancer type and stage. In patients with different advanced stage solid cancers, findings from Phase 1 clinical trials revealed DLT following deescalation after administration of 538 mg/m2/day (Bhargava et al, Clinical Cancer Research, 2001). Data recovered from such early clinical trials indicated that a squalamine dose of about 500 mg/m2/day was well tolerated. Adverse effects included transient and reversible elevation of hepatic transaminases, and PK studies of the plasma concentrations of squalamine showed that circulating levels of squalamine exceeded those which are biologically needed for antitumor activity. In patients with non-small cell 
lung cancers (NSCLC), data from Phase I/IIA trials with squalamine administered in combination with chemotherapeutic agents indicate DLT at 400 mg/m2/day (Herbst et al, Clinical Cancer Research, 2003). In these combination treatment trials, lower doses of squalamine were actually tested, with evidence for minimal adverse side-effects. Further clinical trials are needed to affirm the DLT across different cancer cell types and treatment groups. Importantly, these NSCLC trials provided evidence for objective antitumor responses and clinical benefit. 

In preclinical studies, Li et al. (16) reported that squalamine at 160 nanomolar halted proliferation of human vascular endothelial cells. Remarkably, at such low doses, squalamine significantly reduced VEGF-induced capillary tube-like formations by the endothelial cells growing in 3D cell cultures in vitro. Similarly, in experiments to assess effects of squalamine on phosphorylation of FAK in HUVEC cells, effects of squalamine occurred at doses ranging from 1 nanomolar to 10 micromolar, with maximal inhibition of pFAK at 1 micromolar SQ. In reviewing the data from these experiments MarquezGarban et al. (Marquez-Garban et al, 2019, Cancer Letters), the estimated IC50 for inhibition of pFAK is between 10-100 nanomolar squalamine. Using BxPC3 and Huh7 cell lines, Carmona et al. (Oncotarget, 2019) reported that induction of pancreatic and hepatic cell death in vitro was achieved at IC50 of 5µM and 3µM at 24 and 72 hours, respectively, on dosing with a squalamine derivative. Refer also to Mammari et al. (2022).

Comment: ‘The paragraph about VEGF from line 79 can be moved to the introduction to better fit the paper.

RESPONSE:
Please note that the paragraph listed on line 79 is included in the introduction in the revised manuscript.

Comment: ‘Which type of cancer has an effect when treated with Squalamines and what is the efficacy rate?

RESPONSE: As detailed in the manuscript, several different solid malignancies have been reported to be sensitive to treatment with squalamine across preclinical and clinical trial studies. In preclinical in vivo animal models, squalamine or squalamine derivatives are reported to exhibit strong antitumor efficacy in blocking the progression of: MCF-7 breast cancer cells (Marquez-Garban et al, 2019, Cancer Letters), BxPC-3 and MiaPaCa2 pancreatic cells (Carmona et al. Oncotarget, 2019), HepG2 and Huh7 hepatic cancer cell models (Carmona et al. Oncotarget, 2019), non-small cell lung cancer cell lines that 
are chemotherapy-resistant [66], H460 large cell carcinoma cell lines, NL20T-A and mouse Lewis lung carcinoma [67], human non-small cell lung tumor H23 xenografts [69] and human ovarian cancer cell xenografts (Li et al, Oncogene, 2002). Collectively, the strong evidence of squalamine’s antitumor efficacy in preclinical data led to the initiation of clinical trial studies. Accordingly, results from early phase clinical trials provided evidence that squalamine had antitumor efficacy, as well as a relatively low toxicity profile in advanced solid cancers (Bhargava et al, Clinical Cancer Research, 2001) including NSCLC and ovarian cancers (Herbst et al, Clinical Cancer Research, 2003). In these early clinical trials, squalamine treatment particularly in combination with standard chemotherapy was found to result in promising objective responses and clinical benefit as detailed in the manuscript. 

Concluding Remarks: After in depth review of the manuscript as guided by the reviewer critiques, please note that we have edited some spelling/grammatical errors in the original manuscript text that we corrected in the revised manuscript. These and other modifications requested by reviewers were changed using ‘track changes’ in the revised manuscript in order to accommodate the journal readers.